# REMEDY: Recipe Merging Dynamics in Large Vision-Language Models

**Didi Zhu**[♠,♣]**, Yibing Song**[♣,♠]**, Tao Shen**[♠,♣]**, Ziyu Zhao**[♠,♠]**, Jinluan Yang**[♠,♠]**, Min Zhang**[♦]**, Chao Wu**[♠,♥]*

[♠]Zhejiang University, [♣]DAMO Academy, Alibaba Group, [♦]East China Normal University,
[♥]Academy of Social Governance, Zhejiang University, [♠]Hupan Lab
[♠]Academy of Computer Science and Technology, Zhejiang University
didi_zhu@zju.edu.cn, chao.wu@zju.edu.cn

## Abstract

Model merging has emerged as a powerful technique for combining task-specific vision models into a unified and multi-functional model. Previous methods represented by task arithmetic, have demonstrated effectiveness and scalability in this domain. When large vision-language models (LVLMs) arise with model size scaling up, this design becomes challenging to fuse different instruction-tuned LVLMs for generalization enhancement. The large scale and multi-modal nature of LVLMs present unique obstacles, including constructing reusable and modular components to accommodate the multi-component architecture of LVLMs and the requirement for dynamic fusion based on multi-modal input tokens. To address these challenges, we propose the **RE**cipe **ME**rging **DY**namics (REMEDY) method, a scalable and flexible paradigm for model merging in LVLMs. We first define reusable modules termed *recipes* including the projector and shallow LLM layers, enhancing visual-language understanding. Then, we introduce a modality-aware allocator dynamically generates weights in a one-shot manner based on input relevance to existing recipes, enabling efficient cross-modal knowledge integration. REMEDY thus offers an adaptive solution for LVLMs to tackle both seen (i.e., multi-task learning) and unseen (i.e., zero-shot generalization) tasks. Experimental results demonstrate that our method consistently improves performance on both seen and unseen tasks, underscoring the effectiveness of REMEDY in diverse multi-modal scenarios.

## 1 Introduction

Visual recognition strives to establish a robust alignment between visual perception and linguistic understanding (Liu et al., 2020). It has evolved from basic image classification (Krizhevsky et al., 2012; Huang et al., 2024b; 2023; 2022; Zhang et al., 2024; 2023b; Zhu et al., 2023b; Tong et al., 2023; Zhu et al., 2023c; 2024a; Chen et al.) to complex tasks like detection and segmentation (Anderson et al., 2018; Feng et al., 2022; Bian et al., 2024; Feng et al., 2025; Zhu et al., 2023d; 2024c). This progression has led to advanced research in visual grounding (Plummer et al., 2015), referring expression comprehension (Yu et al., 2016), and referring image segmentation (Hu et al., 2016). Along with model scaling up, the alignment between vision and language has been advanced in the Large Vision-Language Models (LVLMs) form (Lyu et al., 2022a; Dai et al., 2023; Liu et al., 2023; Zhu et al., 2023a; Wang et al., 2023; Zhu et al., 2024b; Lv et al., 2025a; Zhao et al., 2024a). The visual content is projected into the language space for visual question answering (VQA) via large language models (LLM). As the variety and complexity of vision-language (VL) tasks continue to grow, the demand for quickly and efficiently adapting LVLMs to downstream tasks has become increasingly critical.

In light of this, model merging emerges as a promising paradigm, facilitating the synthesis of task-specific models into a unified model capable of simultaneously addressing diverse downstream tasks (Ilharco et al., 2022; Yadav et al., 2023). Prior studies such as the task arithmetic (Ilharco et al., 2022) design develop 'task vectors' for model merging via arithmetic operations, which benefit knowledge

---

*Corresponding author

transfer across different tasks. Building on this, TIES-Merging (Yadav et al., 2023) and AdaMerging (Yang et al., 2024b) advance the merging process to achieve greater versatility by model pruning and adapting merging coefficients, respectively. They have achieved significant success in vision models for recognition scenarios while leaving LVLMs under-explored.

Merging LVLMs encounters emerging challenges, which are analyzed in two aspects. (1) **Large Scale of LVLMs:** LVLM typically consists of a visual encoder, projector, and LLM (Dai et al., 2023; Liu et al., 2023). The parameters of the visual encoder and LLM are typically huge (e.g., in LLaVA 1.5 there are 0.3B vision encoder and 7B or 13B LLM). Adapting these large models to specific tasks is challenging due to limited task-specific data. For instance, the ScienceQA dataset has only 16,96 image-question pairs (Lu et al., 2022), insufficient for comprehensive fine-tuning. *Therefore, focusing on specific LVLM submodules for efficient knowledge transfer becomes crucial.* (2) **Specificity of VL Task Discrepancies:** In LVLMs, the inputs contain both visual and language data, which differs from prior model merging designs where only visual content is processed. Unlike unimodal scenarios, task discrepancies in LVLMs can arise from either the visual or linguistic aspects, or both. As shown in Figure 1, the same globe image might require inferring a city name in one task, while in another task it may need to generate an image description. This variability extends to zero-shot scenarios, where LVLMs are expected to handle novel visual-linguistic combinations, a challenge surpassing traditional unimodal zero-shot tasks. To enhance both multi-task learning and zero-shot generalization, *there is a need for dynamic fusion mechanisms capable of adjusting model merging strategies for each visual-language pair.*

To address the challenges outlined above, we propose **RE**cipe **ME**rging **DY**namics (REM-EDY), a paradigm for merging models in LVLMs that overcomes the constraints of traditional vision-based model merging strategies. As illustrated in Figure 1, REMEDY is composed of two main steps: (1) **Recipe Construction:** We define the reusable modules mentioned in the challenges as *recipes*. Through extensive experiments on LVLMs, we have determined the effective composition of these recipes, which includes the projector and the shallow layers of the large language model. This configuration enhances the model's visual perception capabilities and improves visual-language interaction understanding, rather than merely mimicking output styles (Ghosh et al., 2024). (2) **Recipe Merging:** After constructing the recipes, We introduce a modality-aware allocator and employ a few-shot learning

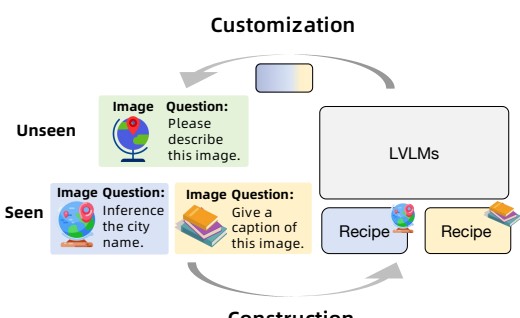

Figure 1: **Construction and Customization Process of REMEDY.** The construction phase uses datasets from "Seen" tasks to create model recipes, while the customization phase applies these recipes to "Unseen" tasks.

paradigm to train it, enabling efficient adaptation with minimal data requirements. This allocator can instantly capture the knowledge relevance between input image-text tokens and existing recipes, and perform one-shot weight allocation based on this captured relevance. By dynamically adjusting to the multi-modal nature of inputs, it facilitates effective integration of knowledge across different tasks and modalities. Results from extensive experiments demonstrate the empirical effectiveness of REMEDY on multimodal benchmarks, showcasing substantial improvements in performance over previous methods.

The main contributions of this work are threefold. First, we analyze LVLM components and construct reusable *recipes* for efficient adaptation. Second, we propose REMEDY with a modality-aware allocator that enables dynamic, one-shot weight allocation through few-shot learning. Finally, through extensive experiments, we demonstrate REMEDY's effectiveness in model merging and knowledge transfer across tasks.

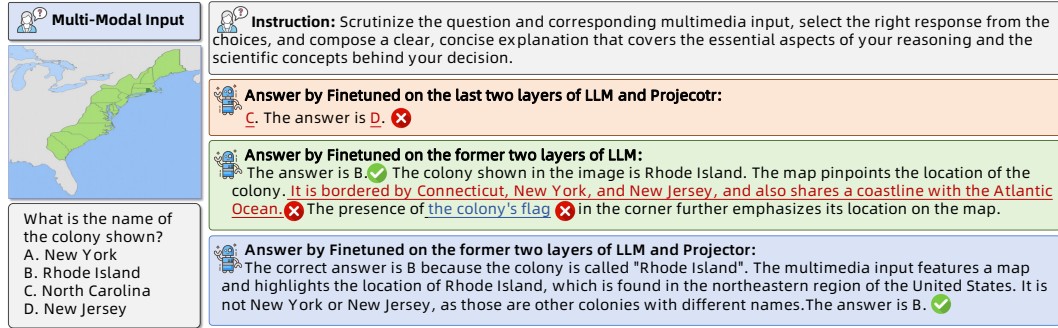

Figure 2: **Comparison of fine-tuning strategies for LLaVA model performance on multi-modal question answering.** Fine-tuning the former two layers yields superior results compared to fine-tuning the last two layers, demonstrating both accurate answers and comprehensive explanations. This suggests that focusing on earlier layers during fine-tuning may be more effective in enhancing LLaVA's comprehension and output quality for multi-modal tasks.

## 2 EXPLORING RECIPE CONSTRUCTION IN LVLMS

Large vision-language models represented by architectures like LLaVA, consist of three core components: a vision encoder, a language model, and a cross-modal fusion module that integrates visual and textual representations. Due to the multi-component nature of LVLMs, we introduce *recipes* as modular functional building blocks for flexible fine-tuning, designed for efficient combination and reuse. These recipes not only enable efficient adaptation to diverse tasks but also reduce computational overhead, which in turn facilitates the subsequent fusion of task-specific knowledge. In developing these recipes for LVLMs, a critical question arises: *which specific components should be fine-tuned to construct an effective recipe?*

To address this, we conducted a comprehensive analysis of LLaVA's structure to identify the key elements that contribute most significantly to task-specific adaptation.

**Shallow Layers vs. Deep Layers.** Our experiments reveal a significant performance disparity when applying Low-Rank Adaptation (LoRA) to shallow versus transformer layers of the LLM. As demonstrated in Table 1, fine-tuning the first two layers of the LLM with a projector consistently achieves performance comparable to, or even surpassing, full model tuning across various datasets. Notably, on the TextVQA dataset, this shallow tuning approach outperforms even the fully-tuned model. In Figure 5, the model fine-tuned on the former two

Table 1: **Performance comparison across different fine-tuning strategies.** Fine-tuning the first two layers with projector achieves comparable or superior performance to full model tuning.

| Dataset | All Layers w/ Proj | First 2 w/ Proj | Last 2 w/ Proj | First 2 w/o Proj |
|---------|--------------------|-----------------|-----------------|------------------|
| ScienceQA | 83.23 | 83.15 | 78.43 | 75.06 |
| Flickr30k | 91.4 | 91.1 | 89.6 | 84.5 |
| COCO | 132.8 | 132.4 | 131.8 | 126.4 |
| TextVQA | 59.77 | 61.89 | 60.39 | 59.34 |

layers provides accurate answers with comprehensive explanations, whereas the model tuned on the last two layers fails to produce correct responses. Specifically, the latter model outputs an incorrect and lack of explanatory context. This behavior likely stems from the format of the training data, where answers may have been presented in a concise, direct manner. As a result, the model appears to merely mimic the output text style without truly comprehending the multi-modal input or the question at hand, aligning with the findings in (Ghosh et al., 2024).

**Effectiveness of Projectors** As illustrated in Table 1, across all datasets, the configuration with projectors consistently outperforms the one without. This performance gap is particularly pronounced in tasks requiring fine-grained visual understanding, such as TextVQA, where the projector-enhanced model surpasses even the fully-tuned model. In Figure 5, neglecting to fine-tune the projector can significantly impair the model's visual perception capabilities. This deficiency may lead to visual hallucinations, such as the erroneous mention of a "flag" that is absent from the image. The impact underscores the critical role of the projector in maintaining the integrity of visual information processing within the model. Based on the above findings, we establish the following guidelines for constructing effective recipes:

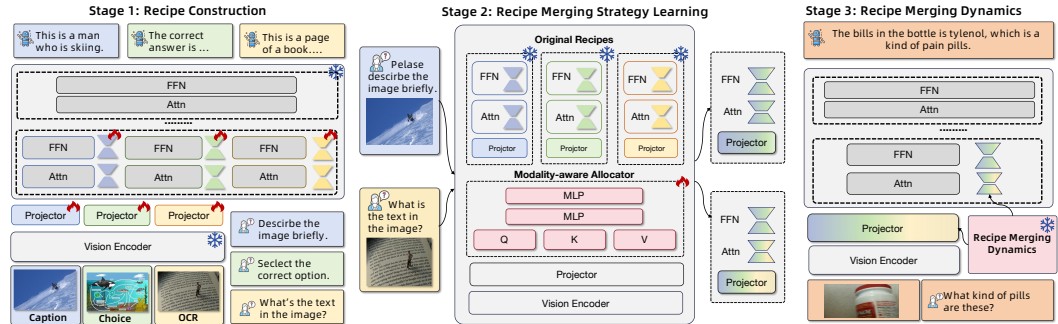

Figure 3: **Overview of Recipe Merging Dynamics for large vision-language models.** Stage 1: We build task-specific recipes using the former layers of LLM and task-specific projectors for different tasks. Stage 2: We employ a modality-aware allocator to learn generalizable merging strategies for the original recipes based on a few seen vision-language tokens. Stage 3: We apply the learned allocator to new multi-modal inputs, dynamically capturing their relationships with existing recipes and assigning layer-wise weights adaptively based on the specific characteristics of each input.

> Fine-tuning the former LLM layers with LoRA and incorporating projectors yields reusable, efficient functional recipes that enhance multi-modal integration and visual understanding.

Building upon our experimental insights, a *recipe* $\mathcal{R}$ for LVLMs is defined as follows:

**Definition 1** (Recipe). *Given an LVLM $\mathcal{M}$ with initial parameters $\boldsymbol{W}$ for its recipe layers in LLM and $\boldsymbol{\Theta}_P$ for its projector, a recipe $\mathcal{R}$ is defined as:*

$$\mathcal{R} = \{\Delta\boldsymbol{W}, \Delta\boldsymbol{\Theta}_P\}, \tag{1}$$

*where $\Delta\boldsymbol{W} \in \mathbb{R}^{M \times N}$ represents the LoRA-induced weight updates for the LLM, with $M$ being the number of adapted Transformer layers, and $N$ being the number of linear layers within each Transformer layer. Each element of $\Delta\boldsymbol{W}$ is defined as $\Delta W_{i,j} = A_{i,j}B_{i,j}^T$, with $A_{i,j}, B_{i,j} \in \mathbb{R}^{d \times r}$ being low-rank matrices and $r \ll d$. $\Delta\boldsymbol{\Theta}_P$ denotes the changes in parameters of the projector.*

For each of the $M$ Transformer layers of the LLM, the updated weight matrices are given by $W'_{i,j} = W_{i,j} + \Delta W_{i,j}$. The projector's updated parameters are $\boldsymbol{\Theta}'_P = \boldsymbol{\Theta}_P + \Delta\boldsymbol{\Theta}_P$. The projector transforms the visual feature vector $v$ by $v' = P(v) = W'_p v + b'_p$, where $W'_p$ and $b'_p$ are components of $\boldsymbol{\Theta}'_P$.

This finding provides valuable guidance for efficient fine-tuning of other LVLMs with similar architectures, as it suggests that focusing adaptation efforts on the initial layers may be sufficient to achieve strong performance while minimizing computational costs.

## 3 METHOD

### 3.1 OBJECTIVE OF RECIPE MERGING

We define two sets of tasks: $\mathcal{T}_{\text{seen}} = \{T_i\}_{i=1}^{S}$, representing the seen tasks on which the model has been fine-tuned through different recipes, and $\mathcal{T}_{\text{unseen}} = \{T_j\}_{j=S+1}^{S+U}$, representing the unseen tasks that have not been encountered during fine-tuning. Each task $T_i$ from the set $\mathcal{T}_{\text{seen}}$ has yielded a corresponding recipe $\mathcal{R}_i = \left(\Delta\boldsymbol{W}^{(i)}, \Delta\boldsymbol{\Theta}_P^{(i)}\right)$, as shown in the left side of Fig. 3.

Our goal is to develop a fusion function, denoted $\mathcal{F}$, that combines all these task-specific recipes into a single set of fused parameters. This fusion function will produce the fused parameters $\boldsymbol{W}^*$ and $\boldsymbol{\Theta}_P^*$:

$$(\boldsymbol{W}^*, \boldsymbol{\Theta}_P^*) = \mathcal{F}\left(\boldsymbol{W}_0, \boldsymbol{\Theta}_P^0, \{\mathcal{R}_i\}_{i=1}^{S}\right), \tag{2}$$

where $\boldsymbol{W}$ and $\boldsymbol{\Theta}_P$ are the initial parameters of the LLM and projector, respectively. It is crucial to note that during the fusion process, only the recipes $\{\mathcal{R}_i\}_{i=1}^S$ are merged, while the remaining parts of the model retain their pre-trained values. The model $\mathcal{M}(\boldsymbol{W}^*, \boldsymbol{\Theta}_P^*)$ is expected to maintain high performance on the tasks in $\mathcal{T}_{\text{seen}}$ while generalizing effectively to the unseen tasks in $\mathcal{T}_{\text{unseen}}$.

We formalize the optimization objective for this fusion as maximizing the combined performance over both seen and unseen tasks:

$$\max_{\boldsymbol{W}^*, \boldsymbol{\Theta}_P^*} \sum_{T_i \in \mathcal{T}_{\text{seen}}} \mathcal{P}\big(\mathcal{M}(\boldsymbol{W}^*, \boldsymbol{\Theta}_P^*), T_i\big) + \sum_{T_j \in \mathcal{T}_{\text{unseen}}} \mathcal{P}\big(\mathcal{M}(\boldsymbol{W}^*, \boldsymbol{\Theta}_P^*), T_j\big), \tag{3}$$

where $\mathcal{P}\big(\mathcal{M}(\boldsymbol{W}^*, \boldsymbol{\Theta}_P^*), T\big)$ represents the performance metric on task $T$ for the given model parameters $(\boldsymbol{W}^*, \boldsymbol{\Theta}_P^*)$.

It is important to note that while our objective function includes performance on unseen tasks, we only have access to data from the seen tasks $\mathcal{T}_{\text{seen}}$ during the training process. This presents a key challenge: how to optimize for generalization to unseen tasks without direct access to their data. Our approach addresses this challenge by leveraging the diverse knowledge captured in the task-specific recipes and designing a fusion mechanism that promotes robust cross-task generalization.

## 3.2 MODALITY-AWARE ALLOCATOR

Building upon the above recipes, we introduce the modality-aware allocator, a mechanism designed to dynamically fuse multiple task-specific recipes. As illustrated in the middle part of Fig. 3, the allocator takes the input token sequence and outputs a single set of dynamic fusion weights for all layers of the recipes.

### 3.2.1 TOKEN UNDERSTANDING COMPONENT.

At the core of the Modality-aware Allocator is the token understanding module, which employs a self-attention mechanism to process and analyze the incoming multimodal token sequences. This module captures complex inter-token relationships and modality-specific relevance, going beyond simple token encoding. Specifically, given an input sequence of tokens $\mathbf{x} = (x_1, x_2, \ldots, x_T)$, where each token $x_t$ may belong to different modalities (e.g., visual or textual), the token understanding module computes modality-sensitive embeddings $\mathbf{h}_t$ for each token: $\mathbf{h}_t = \text{SelfAttention}(x_t, \mathbf{x})$, where $\text{SelfAttention}(\cdot)$ represents the self-attention mechanism that captures dependencies between tokens while accounting for modality information.

By generating these modality-sensitive embeddings $\mathbf{h}_t$, the module provides a nuanced foundation for the subsequent fusion of multiple recipes, allowing the model to adapt its behavior based on the relative importance of visual versus textual information in any given input.

### 3.2.2 TOKEN-CONDITIONAL WEIGHT GENERATOR

Complementing the token understanding module, we introduce the token-conditional weight generator. This component leverages the modality-aware embeddings of the entire input sequence to compute a single set of dynamic fusion weights for all layers of the recipes.

Given the sequence of token embeddings $\{\mathbf{h}_1, \mathbf{h}_2, ..., \mathbf{h}_T\}$ produced by the token understanding module, the weight generator generates a set of weights $\boldsymbol{\gamma} \in \mathbb{R}^{S \times (M \times N + K)}$ in a one-shot manner:

$$\boldsymbol{\gamma} = f_{\text{weight}}(\{\mathbf{h}_1, \mathbf{h}_2, ..., \mathbf{h}_T\}; \boldsymbol{\Theta}_{\text{WG}}), \tag{4}$$

where $f_{\text{weight}}(\cdot)$ is a function parameterized by $\boldsymbol{\Theta}_{\text{WG}}$ (the parameters of the weight generator), $S$ is the number of task-specific recipes, $M$ is the number of Transformer layers, $N$ is the number of linear layers within each Transformer layer, and $K$ is the number of layers in the projector. This function maps the sequence of token embeddings to a single set of weights that determine the contribution of each recipe to each layer of the model. We ensure that the weights are normalized across recipes for each layer by $\sum_{i=1}^S \gamma_{i,j} = 1, \quad \forall j \in \{1, ..., M \times N + K\}$.

**Difference from Routers.** This approach distinguishes itself from conventional mixture-of-experts routers in two key aspects. Firstly, it generates a single set of weights for the entire input sequence in a one-shot manner, rather than producing token-specific weights. Secondly, unlike traditional routers that require deployment at each layer of the model, our method operates globally, generating weights for all layers simultaneously. This holistic, sequence-level weight generation enables more efficient and context-aware processing of multi-modal inputs.

### 3.2.3 TRAINING THE MODALITY-AWARE ALLOCATOR

To train the modality-aware allocator, we randomly sample a small number of image-text pairs from each seen task as training data. Based on the fused parameters defined in Equation 4, we obtain a merged model $\mathcal{M}(\boldsymbol{W}^*, \boldsymbol{\Theta}_P^*)$. Given a training sample $(x, y)$, where $x$ is the input image-text pair and $y = (y_1, ..., y_T)$ is the target text sequence, we define the training objective for the allocator as:

$$\mathcal{L}_{\text{allocator}}(\boldsymbol{\Theta}_{\text{WG}}) = -\sum_{t=1}^{T} \log p(y_t | y_{<t}, x; \boldsymbol{W}^*, \boldsymbol{\Theta}_P^*). \tag{5}$$

By optimizing this objective with respect to only $\boldsymbol{\Theta}_{\text{WG}}$ while keeping all recipe parameters fixed, this autoregressive training objective aligns with the pre-training objective of LVLMs.

### 3.3 RECIPE MERGING DYNAMICS DURING INFERENCE

The modulation of LoRA weights and projector parameters is performed dynamically during inference, as depicted in the right side of Fig. 3. At the start of each forward pass, the input sequence of tokens is processed through the Modality-aware Allocator, which outputs a single set of fusion weights $\gamma$. These weights are then used to fuse the task-specific recipe components. The fused LoRA update $\Delta W_{i,j}^*$ for each layer $(i, j)$ is computed as:

$$\Delta W_{i,j}^* = \sum_{k=1}^{S} \gamma_{k,i,j} \, \Delta W_{i,j}^{(k)}, \quad \forall i \in [1, M], j \in [1, N], \tag{6}$$

where $\gamma_{k,i,j}$ is the fusion weight for the $k$-th recipe at the $i$-th Transformer layer and $j$-th linear layer. Similarly, the fused parameter update $\Delta \Theta_{P,l}^*$ for each layer $l$ of the projector is computed as $\Delta \Theta_{P,l}^* = \sum_{k=1}^{S} \gamma_{k,l} \Delta \Theta_{P,l}^{(k)}, \forall l \in [1, K]$. $\gamma_{k,l}$ is the fusion weight for the $k$-th recipe at the $l$-th projector layer, and $K$ is the number of layers in the projector.

The model's parameters are then updated using these fused components during the forward pass:

$$W_{i,j}' = W_{i,j} + \Delta W_{i,j}^*, \quad \Theta_{P,l}' = \Theta_{P,l} + \Delta \Theta_{P,l}^*. \tag{7}$$

This adaptive fusion mechanism allows the model to effectively leverage knowledge from multiple task-specific recipes, addressing both seen and unseen tasks with improved generalization. By integrating the Modality-aware Allocator into our LVLM, we enable dynamic fusion of multiple recipes, allowing the model to effectively address a diverse set of tasks.

## 4 EXPERIMENT

### 4.1 EXPERIMENT SETUP

**Architectures and Datasets.** To evaluate the effectiveness of REMEDY, we conducted experiments using LLaVA-1.5 (Vicuna-7B), a widely adopted large vision-language model. Our evaluation was performed on two categories of datasets: seen tasks and unseen tasks. Seen tasks refer to datasets that were used during the LLaVA recipe fine-tuning phase to create task-specific recipes. This category includes four datasets: Flickr30k (Young et al., 2014) and COCO (Lin et al., 2014) for image captioning, and ScienceQA (Lu et al., 2022) and TextVQA (Singh

Table 2: **Performance comparison of model fusion methods on seen and unseen tasks.** H-score represents the harmonic mean of the performance on seen and unseen tasks, providing a balanced measure of the model's capability across both task categories. REMEDY demonstrates superior performance compared to other fusion methods and the zero-shot baseline.

| Method | Seen tasks | | | | | Unseen tasks | | | | | | All tasks | |
|---|---|---|---|---|---|---|---|---|---|---|---|---|---|
| | TextVQA | SQA(img) | COCO | Flickr30k | Avg | MM-Vet | MMB-CN | MMB-EN | VizWiz | POPE | Avg | Avg | Hscore |
| Zero-shot | 58.27 | 67.72 | 110.7 | 74.1 | 77.70 | 31.1 | 58.3 | 64.3 | 50.0 | 85.27 | 57.79 | 67.75 | 66.31 |
| Simple Average | 58.81 | 73.89 | 117.8 | 86.6 | 84.28 | 28.4 | 56.34 | 61.3 | 50.8 | 83.42 | 56.05 | 70.17 | 67.31 |
| Task Arithmetic | 57.54 | 72.43 | 115.6 | 84.5 | 82.52 | 27.3 | 55.56 | 60.9 | 49.6 | 82.18 | 55.11 | 68.82 | 66.07 |
| Ties-Merging | 50.45 | 68.45 | 109.2 | 79.1 | 76.80 | 23.8 | 51.19 | 57.3 | 48.6 | 80.32 | 52.24 | 64.52 | 62.21 |
| TW AdaMerging | 59.34 | 72.89 | 114.6 | 82.8 | 82.41 | 27.9 | 54.82 | 59.3 | 50.9 | 83.99 | 55.38 | 68.90 | 66.21 |
| LW AdaMerging | 60.99 | 73.67 | 116.4 | 86.9 | 84.74 | 28.1 | 53.24 | 60.1 | 51.2 | 84.41 | 55.41 | 70.08 | 67.07 |
| REMEDY | 60.34 | 75.34 | 116.9 | 88.2 | 85.20 | 30.9 | 58.69 | 64.9 | 52.2 | 84.88 | 58.31 | 71.76 | 69.25 |

Table 3: **Performance comparison of individual recipes on seen and unseen tasks.** Single-task finetuned recipes demonstrate transferability on a few tasks. REMEDY leverages this transferability to learn dynamic knowledge integration from multiple recipes.

| Method | Seen tasks | | | | | Unseen tasks | | | | | | All tasks | |
|---|---|---|---|---|---|---|---|---|---|---|---|---|---|
| | TextVQA | SQA(img) | COCO | Flickr30k | Avg | MM-Vet | MMB-CN | MMB-EN | VizWiz | POPE | Avg | Avg | Hscore |
| Zero-shot | 58.27 | 67.72 | 110.7 | 74.1 | 77.70 | 31.1 | 58.3 | 64.3 | 50.0 | 85.27 | 57.79 | 67.75 | 66.31 |
| Recipe-TextVQA | 61.59 | 64.85 | 113.2 | 58.5 | 74.54 | 30.2 | 44.12 | 60.4 | 49.34 | 83.55 | 53.52 | 64.03 | 62.23 |
| Recipe-SQA | 53.73 | 83.18 | 100.8 | 61.4 | 74.78 | 28.3 | 57.24 | 65.8 | 49.56 | 83.36 | 56.85 | 65.82 | 64.52 |
| Recipe-COCO | 54.74 | 62.17 | 132.4 | 76.2 | 81.38 | 27.4 | 50.82 | 61.3 | 52.94 | 84.62 | 55.42 | 68.40 | 65.91 |
| Recipe-Flickr30k | 54.58 | 64.40 | 107.9 | 91.1 | 79.50 | 26.5 | 52.24 | 63.2 | 53.57 | 82.54 | 55.61 | 67.56 | 65.44 |
| REMEDY | 60.34 | 75.34 | 116.9 | 88.2 | 85.20 | 30.9 | 58.69 | 64.9 | 52.2 | 84.88 | 58.31 | 71.76 | 69.25 |

et al., 2019) for visual question answering (VQA). Unseen tasks are datasets that were not involved in the recipe construction process. These tasks are used to evaluate the zero-shot generalizability of our method. Specifically, we employed the MM-Vet (Yu et al., 2024), MM-Bench (Zhang et al., 2023a), MM-Bench-Chinese (Zhang et al., 2023a), VizWiz (Gurari et al., 2018), and POPE (Li et al., 2023) datasets to assess this capability. Additionally, we utilize TextCaps (Sidorov et al., 2020), which shares identical images with TextVQA but differs in task instructions, as a special case to analyze REMEDY's allocation behavior across related tasks. For more detailed information about these datasets, please refer to Section A.1 in the Appendix.

**Comparison Methods.** Our comparative analysis encompasses three primary areas: First, we assess the Zero-Shot performance of pre-trained models across various datasets, evaluating their inherent capabilities without any task-specific fine-tuning. Second, we investigate the performance of models fine-tuned on specific datasets, testing these task-specific recipes (Recipe-TextVQA, Recipe-SQA, Recipe-COCO, Recipe-Flickr30k) across

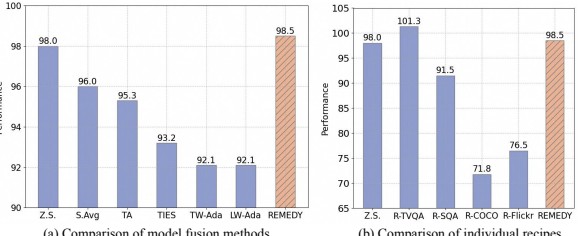

(a) Comparison of model fusion methods    (b) Comparison of individual recipes

Figure 4: **Performance comparison on Related Tasks (TextCaps).** TextCaps shares identical images with TextVQA but differs in task instructions.

all datasets in our study. Third, we compare our method against existing model fusion techniques, including Simple Averaging, Task Arithmetic (Asai et al., 2022), TIES-merging (Yadav et al., 2024), Task-wise AdaMerging (Yang et al., 2023), and Layer-wise AdaMerging (Yang et al., 2023). These techniques aim to combine the strengths of multiple models or efficiently adapt models to new tasks. For more detailed information about these methods and implementation details, please refer to Section A.2 and A.4 in the Appendix.

## 4.2 MAIN RESULTS

Table 2 presents a comprehensive performance comparison between REMEDY and baseline methods across various vision-language tasks, demonstrating REMEDY's superior performance in terms of average performance and H-score. Our experimental results reveal several significant findings:

- **REMEDY demonstrates superior multi-task learning capabilities across seen tasks.** REMEDY significantly outperforms existing model fusion algorithms across seen tasks. This performance was consistently high across diverse seen tasks such as image captioning, visual ques-

tion answering, and visual reasoning. It demonstrates its efficacy in mitigating task interference through dynamic integration. Unlike conventional approaches that often suffer from negative transfer, REMEDY maintains high performance across diverse tasks, indicating successful synergistic knowledge sharing.

- **REMEDY exhibits exceptional zero-shot generalization to unseen tasks.** Notably, current model fusion algorithms generally underperform compared to the simple average method on unseen tasks. This suggests that existing fusion techniques may struggle with generalization, potentially overfitting to the seen tasks. In contrast, REMEDY is the only method that consistently outperforms both the simple average method and the zero-shot baseline across diverse unseen tasks. This improvement demonstrates that REMEDY effectively identifies and leverages relevant cross-modal knowledge for novel scenarios. Specifically, it surpasses the zero-shot performance on MM-Bench and VizWiz datasets without additional training.

- **REMEDY demonstrates effective knowledge transfer between related tasks.** As shown in Table 4, on TextCaps which shares images with TextVQA but has different task objectives, REMEDY outperforms all model fusion methods and individual recipes except Recipe-TextVQA. This indicates that REMEDY can effectively leverage shared visual understanding while maintaining task-specific capabilities, validating our modality-aware allocation strategy for balancing knowledge transfer between related but distinct tasks.

### 4.3 TASK SIMILARITY ANALYSIS

While our main results demonstrate REMEDY's superior performance across both seen and unseen tasks, it is crucial to understand the underlying factors that contribute to knowledge transfer in multi-task learning scenarios.

As illustrated in Table 3, we further analyze the performance of single-task fine-tuned recipes across various tasks. While catastrophic forgetting is prevalent, with many recipes performing worse than zero-shot baselines on out-of-domain tasks, we also observe instances of positive transfer and successful generalization. Several instances demonstrate positive transfer and generalization. Key observations include: (1) Recipe-SQA outperforms zero-shot baselines on MMBench; (2) Recipe-COCO and Recipe-Flickr30k show improved results on VizWiz; (3) Recipe-TextVQA generalizing effectively to TextCaps.

**The effectiveness of this transfer can be attributed to shared characteristics between task domains.** For instance, TextVQA and TextCaps share similar image distributions but differ in their instructions, allowing the recipe to leverage common visual understanding skills. For VizWiz, we observe the Flickr30k recipe receives higher weights - possibly because both tasks require general visual understanding capabilities. Meanwhile, despite both being VQA tasks, VizWiz's unique characteristics (images taken by visually impaired users) may explain why TextVQA recipe receives lower weights. This observation is supported by Table 3, where the Flickr30k recipe shows better transferability to VizWiz (53.57% accuracy).

However, these positive transfer cases are limited to specific task pairs, highlighting the constraints of single-task fine-tuning approaches. **In contrast, REMEDY's dynamic knowledge integration mechanism allows it to effectively leverage transfer opportunities across a much wider range of tasks, both seen and unseen.** This capability results in consistently superior performance and better generalization, underscoring the advantages of REMEDY's approach over single-task recipes.

### 4.4 FURTHER ALLOCATION ANALYSIS

In this subsection, we provide a comprehensive analysis of REMEDY's performance from both qualitative and quantitative perspectives. Figure 5 illustrates four representative examples that highlight the distinctions between our method and simple averaging, along with visualizations of layer-wise allocation scores.

**Qualitative Analysis.** Comparing REMEDY with simple averaging reveals significant differences in output quality across various tasks. For caption tasks, recipe fused by simple averaging tends to produce shorter descriptions, often omitting crucial details. For instance, in the Flickr30k example, it fails to mention "woman" and "picture". In TextCaps, it overlooks the small text on the paper and the phrase "I'm trying to read". REMEDY, in contrast, generates more comprehensive and accurate

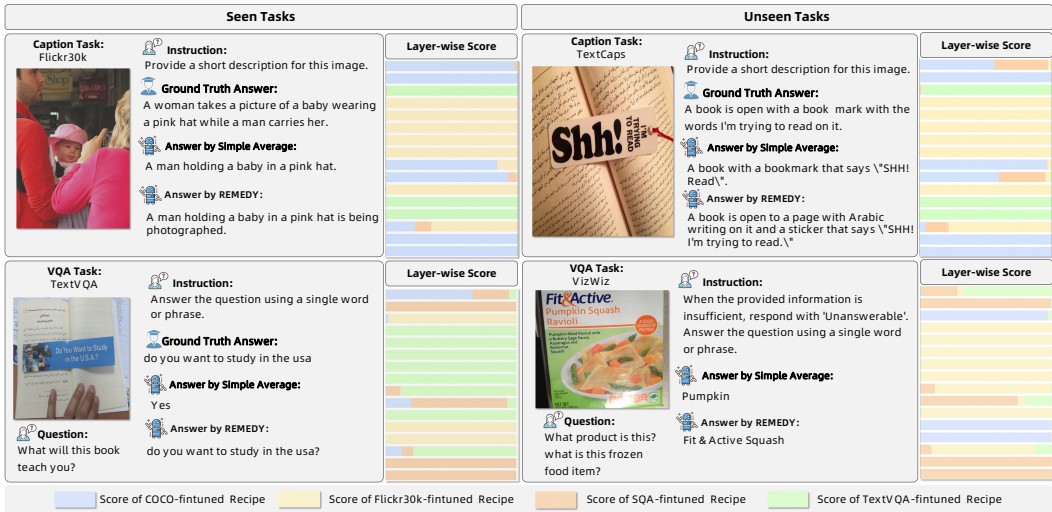

Figure 5: **Comparison and Score Visualization of READY on different multi-modal question answering.** Each row represents a different linear layer, progressing from the projector to the first two sections of the LLM. The colored segments within each row indicate the proportion of scores allocated to different recipes, with segment length corresponding to score magnitude.

captions, capturing these nuanced details. For VQA Tasks, recipe fused by simple averaging exhibits limitations in both visual and textual comprehension. In the TextVQA example, it misinterprets the question, providing an irrelevant answer. For VizWiz, it neglects to address the first part of the question entirely. REMEDY demonstrates superior understanding, providing more accurate and complete responses to these complex queries.

**Quantitative Analysis.** The layer-wise score visualizations in Figure 5 offer insights into REMEDY's allocation mechanism. For caption-related inputs (e.g., Flickr30k and COCO), we observe larger score proportions for the Flickr30k and COCO recipes (yellow and blue segments). TextVQA inputs show higher allocations to SQA-finetuned and TextVQA-finetuned recipes. Interestingly, for the unseen VizWiz dataset, allocations are primarily to Flickr30k-finetuned recipes. This aligns with the observation in Table 3, where Recipe-Flickr30k performed well on VizWiz, demonstrating REMEDY's ability to leverage relevant knowledge from seen tasks for unseen scenarios.

# 5 ABLATION STUDY

**Infulunce of Layer number.** To construct a reuse and effective recipe, we conducted an in-depth investigation into the large language model, a key component of the recipe, focusing particularly on how the number of LoRA fine-tuned transformer layers affects performance. We carried out a series of experiments by adjusting the depth of LoRA fine-tuning, with results presented in Figure 6. The results show that fine-tuning the first 2 layers typically results in substantial performance gains, while extending fine-tuning to deeper layers yields only marginal improvements. This suggests that the initial layers play a crucial role in adapting the model to multi-modal tasks, while fine-tuning additional layers may lead to overfitting. Based on these observation, we selected the strategy of fine-tuning the first 2 layers of the LLM

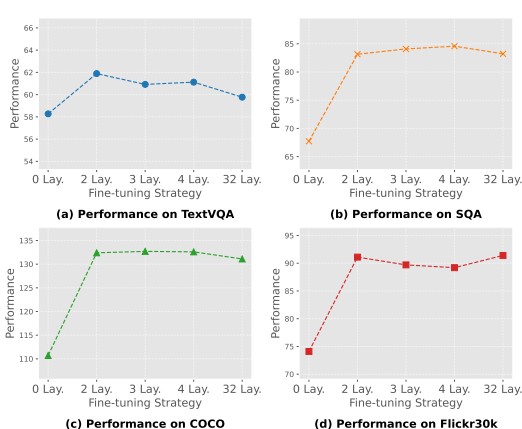

Figure 6: **Impact of fine-tuning different numbers of LLM layers on task performance.**

Table 4: **Performance comparison between REMEDY with attention-based and MLP-based modality-aware allocators on seen and unseen tasks.** The attention-based Allocator consistently outperforms the MLP-based version across all tasks, demonstrating the effectiveness of the attention mechanism in fusing multi-modal information.

| Method | Seen tasks | | | | | Unseen tasks | | | | | | All tasks | |
|---|---|---|---|---|---|---|---|---|---|---|---|---|---|
| | TextVQA | SQA(img) | COCO | Flickr30k | Avg | MM-Vet | MMB-CN | MMB-EN | VizWiz | POPE | Avg | Avg | Hscore |
| REMEDY (MLP) | 58.27 | 67.72 | 115.7 | 85.1 | 81.70 | 28.4 | 55.67 | 62.1 | 50.3 | 83.19 | 55.93 | 68.82 | 66.39 |
| REMEDY | 60.34 | 75.34 | 116.9 | 88.2 | 85.20 | 30.9 | 58.69 | 64.9 | 52.2 | 84.88 | 58.31 | 71.76 | 69.25 |

as the effective trade-off between performance
gain and computational efficiency.

**Effectiveness of Modality-aware Allocator.** We investigated the token understanding component of the modality-aware allocator. Table 4 presents a performance comparison between using attention layers and MLP layers as the token understanding component. In the MLP version, we directly averaged the input tokens before feeding them into the MLP, instead of using an attention mechanism. The results show that the attention-based REMEDY consistently outperforms the MLP-based version across seen and unseen tasks. This consistent improvement demonstrates the crucial role of the attention mechanism in capturing complex relationships between different modalities.

## 6    RELATED WORKS

**Model merging in visual recognition models.** Recent years have witnessed extensive research and application of model merging in visual recognition models. Task Arithmetic (Ilharco et al., 2022) introduced the concept of "task vectors," enabling multi-task learning through arithmetic operations on these vectors. TIES-Merging (Yadav et al., 2023) proposed pruning individual models and eliminating parameter sign conflicts to reduce interference before merging sparse models using Task Arithmetic. AdaMerging (Yang et al., 2024b) presented an adaptive method to learn merging coefficients by minimizing entropy on unlabeled test data as a proxy loss. Representation Surgery (Yang et al., 2024a) observed representation bias in merged models and introduced a "representation surgery" module to calibrate this bias. Consensus (Jain et al., 2023) explored using the Fisher Information Matrix to guide model merging. Additionally, Weight-Ensembling MoE (WEMoE) (Tang et al., 2024) combined model merging with Mixture of Experts (MoE), proposing a dynamic merging Transformer architecture.

**MoE in Large Vision-Language Models.** LLaVA-MoLE (Chen et al., 2024) proposed a sparse mixture of LoRA experts, creating a set of LoRA experts for MLP layers in Transformer layers. This method effectively mitigates conflicts between different instruction datasets while maintaining computational costs. MoCLE (Gou et al., 2024) introduced a cluster-conditional LoRA expert mixture approach. It clusters instructions from all training data, constructs task-specific experts for each cluster, and incorporates a universal expert trained on all data to balance specialization and generalization. TaskGalaxy (Chen et al., 2025) further pushes the boundaries by scaling multi-modal instruction fine-tuning to handle thousands of vision task types Although these methods show promising results in LVLMs, they still have some limitations. These approaches primarily address data conflicts and necessitate concurrent access to all visible task data for joint training of LLM LoRA and routers. However, this requirement may not be feasible in certain practical scenarios. Moreover, the scalability of these methods when applied to large-scale multimodal task sets remains unverified, raising questions about their applicability in more complex, real-world environments.

## 7    CONCLUSION

This paper introduces REMEDY (Recipe Merging Dynamics), a novel approach to model merging for LVLMs. REMEDY addresses the challenges posed by the complex, multi-modal nature of LVLMs, offering a scalable and flexible solution for combining task-specific knowledge. The method consists of two main components: Recipe Construction, which defines reusable modules including the projector and shallow LLM layers, and Recipe Merging, which uses a modality-aware allocator for dynamic weight allocation based on input relevance. This approach enables efficient cross-modal knowledge integration and enhances the model's ability to handle diverse VL tasks.

ACKNOWLEDGMENTS

This work was supported by the National Natural Science Foundation of China (62441605, 62376243), the Zhejiang Provincial Key Research and Development Project (2023C01043), and the Academy Of Social Governance Zhejiang University.

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

# A  DETAILED EXPERIMENT DESCRIPTIONS

## A.1  DETAILED DATASET DESCRIPTIONS

To evaluate the effectiveness of our proposed method, we conducted experiments on a diverse set of vision-language tasks, encompassing both seen and unseen scenarios. Table 5 provides an overview of the datasets used in our study.

**Seen tasks.** For recipe construction and initial evaluation, we utilized four datasets representing different aspects of vision-language understanding: (1) Flickr30K and COCO: Both are image captioning datasets featuring natural scenes. We use CIDEr as the evaluation metric, which captures the consensus between generated and reference captions. (2) SQA(img): A knowledge-grounded visual question answering dataset focused on science topics.It presents multiple-choice questions and is evaluated using accuracy. It's worth noting that the original SQA dataset contains both text-only and image-related questions. However, as we focus on multimodal tasks, we only consider the subset with images, consistent with the approach in Zhou et al. (2024). (3) TextVQA: This dataset tests reading comprehension in visual contexts, requiring models to recognize and understand text within images. Performance is measured by accuracy on phrase-level answers.

**Unseen Tasks.** To assess zero-shot generalization capabilities, we employed five additional datasets not used in recipe construction: (1) MM-Vet: Evaluating the integrated performance of large multimodal models, covering various core visual-language abilities, including recognition, OCR (Optical Character Recognition), knowledge, language generation, spatial perception, and mathematics. (2) MM-Bench: A comprehensive benchmark for multimodal understanding and reasoning, featuring a variety of question types with multiple-choice answers. (3) MM-Bench-Chinese: A Chinese-language subset of the MMBench dataset, designed to evaluate the performance of multimodal models on Chinese cultural and linguistic understanding. (4) VizWiz: This visual question answering dataset is uniquely challenging as it contains images taken by visually impaired individuals, often with quality issues. It requires phrase-level responses. (5) POPE: An object-based VQA dataset designed to probe for object hallucination in vision-language models. It uses yes/no questions to evaluate model performance.

Additionally, we included TextCaps as a related task. While sharing the same image base as TextVQA, this dataset presents a distinct linguistic challenge by requiring the model to generate descriptive captions that incorporate OCR information, rather than answering specific questions. This unique setup allows us to evaluate REMEDY's ability to transfer and combine knowledge across different linguistic objectives.

## A.2  DETAILED COMPARED METHODS

In our study, we compared our proposed approach with several model merging methods. These methods represent different strategies for combining knowledge from multiple tasks or models. Below, we provide a detailed description of each compared method, highlighting their key characteristics and potential advantages:

- **Task Arithmetic**: This method, introduced by Ilharco et al. (2022), involves computing task vectors by subtracting the parameters of a base model from those of a task-specific fine-tuned model. These task vectors can then be added to or subtracted from other models to transfer task-specific knowledge. Task Arithmetic offers a simple yet effective way to combine knowledge from different tasks.

- **TIES-Merging**: Proposed by Yadav et al. (2023), TIES-Merging addresses the challenge of negative transfer in multi-task learning. It employs a task-specific importance estimation strategy to identify and preserve crucial parameters for each task. By pruning less important parameters and aligning the remaining ones, TIES-Merging aims to create a merged model that maintains high performance across all seen tasks.

- **Task-wise AdaMerging**: This approach, developed by Yang et al. (2024b), introduces adaptive merging coefficients at the task level. It dynamically adjusts the contribution of each task-specific model to the final merged model based on the input. This method allows for more flexible knowledge integration, potentially improving performance on diverse inputs by leveraging the most relevant task-specific knowledge for each instance.

Table 5: **Detailed Dataset Descriptions.** "Recipe Construction" indicates whether the dataset was used in the model's recipe finetuning process. This table presents the key characteristics of datasets used in our experiments, including both seen and unseen tasks.

| Dataset | Recipe Construction | Task | Metric | Description | Answer type |
|---|---|---|---|---|---|
| Flickr30K | Yes | Image Caption | CIDEr ($\uparrow$) | Image dataset with captions for natural scenes. | Caption |
| COCO | Yes | Image Caption | CIDEr ($\uparrow$) | Image dataset with captions for natural scenes. | Caption |
| SQA(img) | Yes | Knowledge Grounded VQA | Accuracy ($\uparrow$) | Science-focused multiple-choice VQA | Option |
| TextVQA | Yes | Reading Comprehension VQA | Accuracy ($\uparrow$) | VQA with text recognition in images. | Phrase |
| MM-Vet | No | General VQA | Accuracy ($\uparrow$) | Evaluating the integrated performance of large multimodal models. | Sentence |
| MM-Bench | No | General VQA | Accuracy ($\uparrow$) | Comprehensive benchmark for multimodal understanding and reasoning. | Option |
| MM-Bench-Chinese | No | General VQA | Accuracy ($\uparrow$) | Comprehensive benchmark for Chinese cultural and linguistic understanding. | Option |
| VizWiz | No | General VQA | Accuracy ($\uparrow$) | VQA dataset focused on images taken by visually impaired individuals. | Phrase |
| POPE | No | Object-based VQA | Accuracy ($\uparrow$) | Probing dataset for object hallucination in vision-language models. | Yes/No |
| TextCaps | No | Image Caption | CIDEr ($\uparrow$) | Image dataset with captions for natural scenes. | Caption |

Table 6: **Recipe Finetuning Configurations**

| Dataset | Data Size | Learning Rate | Training Epochs | Training Instruction |
|---|---|---|---|---|
| Flickr30K | 145,000 | 2e-5 | 1 | Provide a one-sentence caption for the provided image. |
| COCO | 113,287 | 2e-5 | 1 | Provide a one-sentence caption for the provided image. |
| SQA | 16,967 | 2e-4 | 5 | Answer with the option's index (start from A) from the given choices directly. |
| TextVQA | 34,602 | 2e-5 | 5 | Answer the question using a single word or phrase. |

- **Layer-wise AdaMerging**: An extension of the AdaMerging concept, this method applies adaptive merging at the layer level rather than the task level. It allows for finer-grained control over the merging process, capturing layer-specific expertise from different models.

## A.3 EVALUATION METRICS

To comprehensively evaluate the performance of our method, we employ two primary metrics: **H-score** and **Avg**. H-score represents the harmonic mean of the performance on seen and unseen tasks, providing a balanced measure of the model's capability across both task categories. Avg represents the average performance across all tasks.

## A.4 DETAILED IMPLEMENTATION DETAILS

For the recipe construction phase, we focus on two key components of the LLaVA architecture. First, we target the initial two layers of the LLM component. For fine-tuning the LLM layers, we employ the LoRA technique. The second component we fine-tune is the projector. For this part, we utilize standard fine-tuning techniques, allowing for a more comprehensive update of the projection parameters. For the recipe merging phase, we train a modality-aware allocator by a standard fine-tuning process.

**Implementation Details of Recipe Construction.** Table 6 shows the configurations used in recipe construction. For image captioning tasks (Flickr30K and COCO), we employed a conservative approach with a low learning rate (2e-5) and a single training epoch, given the large dataset sizes and

Table 7: **Computational analysis of different fusion methods.** While REMEDY requires moderately more computational resources than baseline methods, it achieves significantly better zero-shot performance, demonstrating a favorable trade-off between computational cost and model capability.

| Method | Training (h) | Inference (ms/query) | Training GPU Memory (GB) | Inference GPU Memory (GB) | GFLOPs |
|---|---|---|---|---|---|
| Simple Average | - | 89 | - | 38.5 | 380 |
| LW AdaMerging | 3.84 | 89 | 52.8 | 38.5 | 425 |
| REMEDY | 5.25 | 123 | 60.6 | 40.3 | 483 |

the relatively straightforward nature of the task. In contrast, for more complex tasks like SQA(img) and TextVQA, we adjusted our strategy to account for the increased task difficulty and smaller dataset sizes. This involved using a higher learning rate (2e-4) for SQA(img) and increasing the number of training epochs to 5 for both SQA(img) and TextVQA. These adjustments allowed for more thorough learning on these challenging tasks without risking overfitting. The rank of LoRA is set to 128.

**Implementation Details of Recipe Merging.** The modality-aware allocator is implemented as a neural network module that dynamically adjusts the contribution of different modalities. It consists of an input projection layer implemented by a self-attention mechanism, and a multi-layer perceptron with normalization and dropout. The learning rate of the allocator is set to 2e-4. For training the modality-aware allocator, we randomly sampled 1000 data points from each of the four seen datasets. This sample size represents approximately 1% to 5% of the original training sets, providing a diverse yet manageable subset for efficient training of the allocator. The training objective is to minimize task-specific losses of the merged model on the sampled data. Specifically, we aim to find optimal merging weights generated by the allocator such that when applied to fuse the original recipes, the resulting merged model achieves minimal loss on each task.

## A.5 COMPUTATIONAL ANALYSIS

Table 7 presents a detailed analysis of computational costs and performance metrics for different model fusion approaches. We evaluate these methods across multiple dimensions including training time, inference latency, memory requirements, and computational complexity (GFLOPs).

These results REMEDY introduces moderate additional costs while achieving substantial performance gains. Specifically, the training phase requires about 1.5 more hours compared to LW AdaMerging, mainly due to the modality-aware allocator training. During inference, REMEDY adds a marginal latency per query for weight computation. Given the significant performance improvement (+3.17% over simple averaging on unseen tasks), we believe these computational trade-offs are well justified.

## B MORE RELATED WORKS

**Model merging in LLMs.** LoraHub (Huang et al., 2024a) first trains several LoRA weights on upstream tasks, then uses a gradient-free method to search for optimal combination coefficients for downstream tasks. LoRA-Flow (Wang et al., 2024) introduces fusion gates at each Transformer layer, generating weights to average the outputs of pre-trained LoRAs. Arrow (Ostapenko et al., 2024) explores building and reusing a library of LoRA experts for zero-shot task generalization, adding linear routers at each layer for dynamic selection using expert prototypes. PHATGOOSE (Muqeeth et al., 2024) proposes a slightly modified training procedure, adding sigmoid gates for each training task to learn token importance. LoraRetriever and its variant (Zhao et al., 2024c;b) train a sentence embedding model to map input queries to an embedding space for expert model selection and routing. Recent works have explored rank-wise operations for LoRA fusion by treating each rank as a separate module (Zhao et al., 2024d; 2025). Security concerns have also been addressed with approaches focusing on mitigating backdoor effects and balancing safety in merged models (Yang et al., 2024c; 2025; Lyu et al., 2023; 2022b; 2024b;a). Meanwhile, efficiency improvements have been achieved through sensitivity-guided parameter balancing and dynamic quantized merging strategies (Liu et al., 2025a;b). Lv et al. (2025b) demonstrates the effectiveness of collabo-

rative approaches between large language models and specialized small models in recommendation scenarios.

While these methods show impressive performance in LLMs, they have some limitations. Firstly, they are primarily designed for pure text LLMs and cannot be directly applied to Multimodal Large Language Models (LVLMs). They don't consider the interaction between image and text tokens, which is crucial in LVLMs. Secondly, many methods adopt token-wise fusion strategies and deploy routers at every layer, a design that might lead to computational inefficiency and increased model complexity in LVLMs.

## C   LIMITATIONS AND FUTURE WORK

While REMEDY shows improved zero-shot generalization, there is still room for further enhancement in this area. Additionally, the current recipe construction process relies heavily on experimental findings, which may limit its adaptability to new scenarios. Future work could focus on developing more dynamic recognition mechanisms for recipe construction, potentially incorporating adaptive techniques that can automatically identify effective module compositions based on task characteristics. This could lead to even more flexible and generalizable model merging strategies for LVLMs, further advancing their capabilities in handling diverse and novel multi-modal tasks.

## D   ETHICAL STATEMENT

We declare that our research does not present any potential ethical issues. The study does not involve human subjects, sensitive data, or methodologies that could result in harmful outcomes or biases. All data this work uses is publicly available, and no privacy or security concerns are implicated.

## E   REPRODUCIBILITY STATEMENT

We have made significant efforts to ensure the reproducibility of our work. A code example is provided as supplementary material, demonstrating the core components of our approach. Upon acceptance, we will release all of the data and the complete training and testing code to facilitate the full reproducibility of our results.

