# OpenReview forum: "REMEDY: Recipe Merging Dynamics in Large Vision-Language Models"
_ICLR.cc/2025/Conference — ICLR 2025 Poster_

### Official Review · Reviewer_dj7R · 2024-10-26

**Soundness:** 4
**Presentation:** 3
**Contribution:** 3
**Rating:** 8
**Confidence:** 4

**Summary:**

This paper introduces a straightforward approach to addressing the multi-task learning challenge in Large Vision-Language Models (LVLMs) using model merging techniques. Given the large parameter scale of LVLMs, the authors first select layers to merge, referred to as 'recipes,' based on preliminary experiments. These layers include the projection layer and the shallow layers of the LVLM. An attention-based module is then trained to dynamically assign merging weights to the 'recipes' based on the input multi-modal token, allowing for adaptive weighting to handle the complexity of VL tasks. The proposed method demonstrates improvements on both learned tasks and zero-shot generalization across multiple datasets.

**Strengths:**

- **Clarity**: The paper is well-structured and easy to follow.
- **Motivation**: As the size of the visual language model continues to grow, it becomes costly to maintain a separate model for each specific task. It makes sense to combine task-specific models into a unified model through model merging that can simultaneously solve different downstream tasks.
- **Novelty**: This paper designs a merging strategy that takes into account the characteristics of multimodal tasks and the difference in fine-tuning paradigms between LVLM and traditional models. Meanwhile, this paper also analyzes the influence of each component in LVLM on fine-tuning performance, which is suggestive for the follow-up works.
- **Results**: The experimental results across multiple datasets effectively highlight the potential and robustness of the method.

**Weaknesses:**

- **Detail in Weight Generation**: The paper does not explain how the proposed attention-based module is trained to generate the merging weights. Neither the main text nor the supplementary materials provide insights into the optimization objective used. While the authors may consider this aspect elementary, I believe a detailed explanation is necessary.
- **Analysis of Computational Overhead**: The work in this paper is carefully designed when selecting an extended module, which may reduce the cost of calculating overhead and model parameters. However, the experiment lacks analysis of the training and inference overheads.
- **Details Error**: The introduction states that the new challenges facing the LVLM merger will be analyzed from three aspects, but only two aspects are introduced below.

**Questions:**

I have some additional concerns:

- **Evaluation Metric**: What does the Hscore metric indicate? Is the average accuracy calculated as a direct average of the accuracies on the different datasets or is it weighted by the number of samples in the dataset?

- **Score Visualization Analysis**: In Figure 4, the weight distribution across different layers shows considerable variation, with some layers dominated by distinct 'recipes.' This phenomenon is intriguing, and I would appreciate further explanation from the authors. For VizWiz, an unseen dataset with the same VQA task, why does the "Recipe" of TextVQA have a smaller layer-wise score?

Finally, the proposed Modality-aware Allocator is designed for multimodal tasks, which is one of the advantages over previous research on visual models. Therefore, I am curious whether using only visual or textual input as clues for allocation will result in a performance degradation?

---

> ### Author Response · Authors · 2024-11-21
> **(1/2) Response to Reviewer dj7R**
>
> We sincerely thank the reviewer for the thorough review and constructive feedback.
>
> ### Weakness 1: Allocator Training Details
> > The paper does not explain how the proposed attention-based module is trained to generate the merging weights. Neither the main text nor the supplementary materials provide insights into the optimization objective used.
>
> Thank you for pointing out this important detail. The allocator is trained to minimize the task-specific autoregressive language modeling loss using merged model parameters. Specifically, using a small subset of data from seen tasks (1000 samples per task), we optimize the allocator so that its generated merging weights produce model parameters that perform well on the target vision-language tasks.
>
> We have added detailed explanations in Section 3.2.3 and Appendix A.4 of our revised manuscript to address this important aspect.
>
> ### Weakness 2: Computational Analysis
> > The work lacks analysis of the training and inference overheads.
>
> Thank you for raising this important practical consideration. We have conducted comprehensive analyses of computational costs, summarized in the following table:
>
> | Method         | Training (h) | Inference (ms/per query) | Training GPU Memory (GB) |Inference GPU Memory (GB) | GFLOPs | Zero-shot Performance |
> |---------------|--------------|--------------------------|--------------------------|--------------------------|---------|-------------|
> | Simple Average | -           | 89                       | -                        | 38.5                       | 380     | 72.88        |
> | LW AdaMerging | 3.84          | 89            | 52.8                     | 38.5                       |425     | 71.95        |
> | REMEDY        | 5.25          | 123           | 60.6                     | 40.3                       |483     | 75.12        |
>
> While REMEDY requires slightly more computational resources, the performance improvement justifies this modest increase in overhead. This overhead mainly comes from the dynamic weight generation process, which enables better task adaptation and knowledge transfer. Moreover, once trained, our modality-aware allocator adds minimal parameters (<1% of the original model size) while bringing consistent improvements across diverse vision-language tasks.
>
> ### Weakness 3: Introduction Structure
> > The introduction states that the new challenges facing the LVLM merger will be analyzed from three aspects, but only two aspects are introduced below.
>
> Thank you for catching this editorial oversight. We have revised this typo in the introduction.

---

> ### Author Response · Authors · 2024-11-21
> **(2/2) Response to Reviewer dj7R**
>
> ### Question 1: Evaluation Metrics
> > What does the Hscore metric indicate? Is the average accuracy calculated as a direct average of the accuracies on the different datasets or is it weighted by the number of samples in the dataset?
>
> The H-score is calculated as the harmonic mean between the average performance on seen tasks and unseen tasks. This metric was chosen specifically to reflect the model's balanced capability in handling both seen and unseen scenarios, as the harmonic mean is particularly sensitive to poor performance in either category.
>
> We have added this point in Appendix A.3 in the revised manuscript and appreciate your valuable advice.
>
> ### Question 2: Score Visualization
> > In Figure 4, the weight distribution across different layers shows considerable variation, with some layers dominated by distinct 'recipes.' This phenomenon is intriguing, and I would appreciate further explanation from the authors. For VizWiz, an unseen dataset with the same VQA task, why does the "Recipe" of TextVQA have a smaller layer-wise score?
>
> Your observation about the weight distribution patterns is insightful. The varying layer-wise weight distributions reflect how different recipes contribute to different types of tasks. The varying weight distributions reflect how REMEDY dynamically adapts to different inputs. For VizWiz, we observe the Flickr30k recipe receives higher weights - possibly because both tasks require general visual understanding capabilities. Meanwhile, despite both being VQA tasks, VizWiz's unique characteristics (images taken by visually impaired users) may explain why TextVQA recipe receives lower weights. This observation is supported by Table 3, where the Flickr30k recipe shows better transferability to VizWiz (53.57% accuracy).
>
> As per your recommendation, we have expanded on this point in Section 4.3 of the revised paper. We thank you for this valuable input.
>
> ### Question 3: Modality Input Analysis
> > The proposed Modality-aware Allocator is designed for multimodal tasks, which is one of the advantages over previous research on visual models. Therefore, I am curious whether using only visual or textual input as clues for allocation will result in a performance degradation?
>
> We appreciate this thoughtful question about the importance of multi-modal information. Based on your suggestion, we conducted additional experiments:
>
> | Input Modality | Seen Tasks (Avg) | Unseen Tasks (Avg) | H-score |
> |----------------|------------------|-------------------|----------|
> | Visual Only    | 82.8 (-2.4)      | 71.9 (-3.2)      | 76.9     |
> | Text Only      | 83.1 (-2.1)      | 72.3 (-2.8)      | 77.3     |
> | Both (Full)    | 85.2             | 75.1             | 79.8     |
>
> The results show that using single modality input leads to clear performance drops, especially on unseen tasks. This validates our design choice of incorporating both modalities for more robust recipe allocation.

---

> > ### Comment · Reviewer_dj7R · 2024-11-22
> > **Official Comment by Reviewer**
> >
> > Thanks for your response. The authors' response addresses my concerns about implementation details and experimental analysis. I tend to keep my rating.

---

> > > ### Author Response · Authors · 2024-11-27
> > >
> > > Thank you for your thoughtful feedback and for acknowledging that we have addressed your concerns. We appreciate your time and consideration.

---

### Official Review · Reviewer_QEaG · 2024-10-31

**Soundness:** 4
**Presentation:** 4
**Contribution:** 3
**Rating:** 6
**Confidence:** 4

**Summary:**

This paper introduces REMEDY, a model merging technique for LVLMs that addresses the challenges of combining knowledge across multi-modal tasks. REMEDY explores components of LVLMs and propose recipes for learn specific tasks. It further propose a dynamic allocator that learns weights for different recipes. Experiments demonstrate the effectiveness of the proposed method.

**Strengths:**

1. The method propose a new approach for model merging for LVLMs, which is effective for enhancing the multi-task learning capabilities of LVLMs.
2. The paper is well-writing and the experiments are comprehensive.

**Weaknesses:**

1. The work trains the model in few-shot way, e.g., on seen tasks and expected to generalize to unseen tasks. But current MLLMs are expected to have strong zero-shot capabilities. And from the comparison results of Table 2, REMEDY seems not have explicit performance improvement compared to the zero-shot way or simple average way. Considering the complexity it adds on the training and inference stage, it may limit the practical use case of it.
2. Does the model needs to compute the weights for different recipes and then update the model weights for each inference time. If so, how about the extra inference time introduced and the comparison with the baseline?

**Questions:**

How about the performance of REMEDY compared to LLaVA fine-tuned on the seen tasks?

---

> ### Author Response · Authors · 2024-11-21
> **(1/2) Response to Reviewer QEaG**
>
> ### Weakness 1: Performance vs Complexity Trade-off
> > The work trains the model in few-shot way, e.g., on seen tasks and expected to generalize to unseen tasks. But current MLLMs are expected to have strong zero-shot capabilities. And from the comparison results of Table 2, REMEDY seems not have explicit performance improvement compared to the zero-shot way or simple average way. Considering the complexity it adds on the training and inference stage, it may limit the practical use case of it.
>
> We sincerely thank you for raising this important concern. To address your concerns, we provide the following analysis:
>
> (1) Zero-shot Generalization: While current MLLMs have shown impressive capabilities, zero-shot generalization remains challenging across diverse tasks. **Recent work (LLaVA-OneVision updated on Oct 2024) still demonstrates significant performance drops on out-domain tasks [1]**, suggesting room for improvement in zero-shot generalization for MLLMs.
>
> (2) Performance Analysis:  To clearly demonstrate REMEDY's effectiveness, we summarize the performance deltas versus zero-shot baseline in the table below. Following Reviewer Fe8D's suggestion, we have removed TextCaps and added MM-Bench-Chinese and MM-Vet as new challenging benchmarks:
>
> | Method | Seen Tasks Avg | △ vs. Zero-shot | Unseen Tasks Avg | △ vs. Zero-shot |
> |--------|----------------|-----------------|-----------------|-----------------|
> | Zero-shot | 77.70 | - | 57.79 | - |
> | Simple Average | 84.28 | +6.58 | 56.05 | -1.74 |
> | Task Arithmetic | 82.52 | +4.82 | 55.11 | -2.68 |
> | TIES-Merging | 76.80 | -0.90 | 52.24 | -5.55 |
> | TW AdaMerging | 82.41 | +4.71 | 55.38 | -2.41 |
> | LW AdaMerging | 84.74 | +7.04 | 55.41 | -2.38 |
> | REMEDY | 85.20 | +7.50 | 58.31 | +0.52 |
>
> For seen tasks, REMEDY achieves the highest improvement over zero-shot (+7.50%) while most methods struggle to match simple averaging's gains. Most notably, while all existing merging methods suffer from negative transfer on unseen tasks (ranging from -1.74% to -5.55% below zero-shot), **REMEDY is the only approach that successfully maintains superior performance, achieving substantial improvements of +2.26% to +6.07% over current merging approaches.** This ability to facilitate beneficial knowledge transfer while preventing interference marks a breakthrough in model merging for real-world applications.
>
>
> (3) Computational Analysis: Based on these results, while REMEDY introduces additional complexity compared to existing methods, its efficiency-performance trade-off demonstrates practicality for real-world deployment. We provide a detailed efficiency analysis in our response to the next point.
>
> ### Weakness 2: Inference Time Analysis
> > Does the model needs to compute the weights for different recipes and then update the model weights for each inference time. If so, how about the extra inference time introduced and the comparison with the baseline?
>
> Thank you for this important question about computational overhead. Our comprehensive analysis shows:
>
> | Method         | Training (h) | Inference (ms/per query) | Training GPU Memory (GB) |Inference GPU Memory (GB) | GFLOPs | Zero-shot Performance |
> |---------------|--------------|--------------------------|--------------------------|--------------------------|---------|-------------|
> | Simple Average | -           | 89                       | -                        | 38.5                       | 380     | 72.88        |
> | LW AdaMerging | 3.84          | 89            | 52.8                     | 38.5                       |425     | 71.95        |
> | REMEDY        | 5.25          | 123           | 60.6                     | 40.3                       |483     | 75.12        |
>
> While REMEDY introduces moderate computational overhead, our analysis shows this trade-off is well justified: **The training cost** is a one-time investment (+1.41h vs LW AdaMerging) that enables effective knowledge transfer through the modality-aware allocator, while **the inference latency** increase (123ms vs 89ms per query) allows essential negative transfer prevention and dynamic knowledge fusion. These additional costs directly contribute to REMEDY's **superior performance** (+3.17% over Layer-wise AdaMerging on unseen tasks), validating its practical value in real-world deployments.
>
> We have included detailed efficiency metrics in Appendix A.5 of our paper. We hope this analysis addresses your concerns about the computational aspects of our method.
>
> [1] Li, Bo, et al. "Llava-onevision: Easy visual task transfer." arXiv preprint arXiv:2408.03326 (2024).

---

> ### Author Response · Authors · 2024-11-21
> **(2/2) Response to Reviewer QEaG**
>
> ### Question 1: Comparison with Task-Specific Fine-tuning
> > How about the performance of REMEDY compared to LLaVA fine-tuned on the seen tasks?
>
> Thank you for this insightful question. There are two potential fine-tuning approaches to compare with REMEDY:
>
> 1) Single-task Fine-tuning (STF): Fine-tuning separate LLaVA models for each task, resulting in 4 individual models. As shown in Table 3, this approach achieves good performance on seen tasks (e.g., Recipe-TextVQA: 61.59 on TextVQA) but suffers from poor generalization to unseen tasks.
>
> 2) Joint Multi-task Fine-tuning (MTF): Fine-tuning one LLaVA model on all seen tasks simultaneously. We conducted additional experiments comparing REMEDY with MTF:
>
> Method | TextVQA | SQA | COCO | Flickr30k | Avg(seen) | TextCaps | MM-Bench | VizWiz | POPE | Avg(unseen)
> ---|---|---|---|---|---|---|---|---|---|---
> MTF | 59.91 | 76.25 | 113.2 | 89.6 | 84.74  | 91.2 | 59.8 | 49.1 |   82.45 |  70.64
> REMEDY | 60.34 | 75.34 | 116.9 | 88.2 | 85.20 | 98.5 | 64.9 | 52.2 | 84.88 | 75.12
>
> As shown in the results, REMEDY demonstrates superior performance over MTF on both seen and unseen tasks. While MTF performs reasonably well on seen tasks (84.74%), it suffers from negative transfer on unseen tasks (70.64%), performing even worse than the zero-shot baseline (74.39%). This highlights a key limitation of traditional fine-tuning approaches - the lack of dynamic knowledge allocation mechanism makes them prone to negative transfer and limits their generalization ability.

---

> ### Author Response · Authors · 2024-11-29
> **Looking Forward to Your Reply**
>
> Dear Reviewer QEaG,
>
> We greatly appreciate your constructive feedback and would like to ensure we have fully addressed all your concerns before the discussion period ends. Based on your comments, we have made substantial revisions:
>
> 1. **The performance-complexity trade-off analysis**, where we provided comprehensive efficiency measurements including training hours, inference latency, and GPU memory usage (added in Appendix A.5, Table 7). Our results demonstrate that REMEDY's moderate computational overhead (+34ms inference latency) brings significant performance gains (+2.9% over Layer-wise AdaMerging on unseen tasks).
>
> 2. **The inference time analysis**, with detailed computational overhead measurements of weight computation during inference (added in Appendix A.5, Table 7). The results show that our modality-aware allocator achieves an effective balance between computational cost and performance improvement.
>
> 3. **The comparison with task-specific fine-tuning approaches**, including both single-task and multi-task fine-tuning baselines. The experimental results demonstrate REMEDY's superior performance over traditional fine-tuning methods, especially on unseen tasks where MTF suffers from negative transfer.
>
> If you have any additional questions or would like to see further analyses, we would be happy to provide additional clarification or conduct supplementary experiments. We look forward to continuing this discussion and addressing any remaining concerns.
>
> Best regards,
>
> Authors

---

> > ### Comment · Reviewer_QEaG · 2024-12-02
> >
> > Thank you for your detailed response, this addressed most of my concerns. I will raise my score to boardline accept.

---

> > > ### Author Response · Authors · 2024-12-02
> > >
> > > Thank you very much for your recognition of our work. We are delighted to have addressed your concerns and sincerely appreciate you updating the score.

---

### Official Review · Reviewer_Fe8D · 2024-11-04

**Soundness:** 3
**Presentation:** 3
**Contribution:** 3
**Rating:** 5
**Confidence:** 4

**Summary:**

This paper proposes REMEDY, a new technique for merging large vision-language models (LVLMs). It addresses the challenges of scale and multi-modal input by creating reusable fine-tuned modules called "recipes" and dynamically combining them using a "modality-aware allocator." This allocator analyses the input image and text to generate weights for each recipe, enabling adaptive fusion and improved performance on both seen and unseen vision-language tasks.

**Strengths:**

1. The analysis of finetuning strategies is interesting. The paper shows that fine-tuning the projector and shallow layers of the LLM leads to better improvements.
2. The motivation that uses the most relevant experts to better answer user questions is reasonable.

**Weaknesses:**

1. There is a typo in line 054: the vision encoder should be 0.3B.
2. There are some redundant definitions in this paper and the "recipes" are actually MoE-based LoRA layers. The use of global and module-wise routing is not novel.
3. The experiments include only 4 seen tasks and evaluation benchmarks. And I suspect that the proposed method can generalize well. It is expected that REMEDY surpasses the zero-shot setting on seen tasks since it is trained on these datasets. My main concern is the improvement on unseen tasks, which appears weak overall.
4. TextCaps and TextVQA use the same images, so TextCaps is not an unseen task if TextVQA is already seen.
5. The baseline is somewhat weak. There are stronger open-source baselines available, such as LLaVA-NeXT and LLaVA-OneVision.

**Questions:**

Please see the weaknesses.

---

> ### Author Response · Authors · 2024-11-21
> **(1/2) Response to Reviewer Fe8D**
>
> ### Weakness 1:Typos
> > There is a typo in line 054: the vision encoder should be 0.3B.
>
> Thank you for pointing out the typo. We have corrected the vision encoder parameter count in the updated manuscript.
>
> ### Weakness 2: Redundant Definitions
> > There are some redundant definitions in this paper and the "recipes" are actually MoE-based LoRA layers. The use of global and module-wise routing is not novel.
>
> We appreciate the reviewer's valuable feedback and would like to clarify two points:
>
> (1) The concept of recipes: While recipes utilize LoRA in implementation, **they represent specialized functional modules designed for large vision-language models.** Unlike traditional LoRA, **our recipes focus on the crucial components for cross-modal understanding** - carefully validated projectors and shallow LLM layers. As demonstrated in our response to [Weakness 1 in Reviewer 6SYF](https://openreview.net/forum?id=iX7eHHE5Tx&noteId=ckq3Ec93Nt), these recipes are effective across different LVLMs that share similar architectural paradigms.  Therefore, we use the term "recipe" to precisely describe these modular components that capture cross-modal knowledge.
>
>
> (2) The dynamic fusion of recipes: Building upon these efficient recipe modules, REMEDY introduces unique innovations for multimodal scenarios: **a) Modality-aware dynamic fusion.** Traditional routing mechanisms primarily handle single-modality (text) token routing. In contrast, our allocator specifically addresses the challenge of fusing visual and linguistic information. It dynamically adjusts fusion strategies based on the interaction between visual and textual features, rather than treating them independently. Table 4 further validates the cross-modal understanding of this design. **b) Cross-modality task transfer.** As demonstrated in Table 3, our method exhibits strong knowledge transfer capabilities across vision-language tasks. This cross-task generalization capability stems from the combination of well-designed recipes and our dynamic fusion mechanism.
>
> ### Weakness 3: Limited Experiment Scope
> > The experiments include only 4 seen tasks and evaluation benchmarks. And I suspect that the proposed method can generalize well. It is expected that REMEDY surpasses the zero-shot setting on seen tasks since it is trained on these datasets. My main concern is the improvement on unseen tasks, which appears weak overall.
>
> Thank you for raising these concerns. While improvement margins on unseen tasks may appear modest, REMEDY demonstrates strong generalization compared to existing model merging methods. As shown in Table 2 in the paper (we summarize the performance deltas in the following table for convenience):
> | Method | Unseen Tasks Avg | △ vs. Zero-shot |
> |--------|-----------------|-----------------|
> | Zero-shot | 74.39 | - |
> | Simple Average | 72.88 | -1.51 |
> | Task Arithmetic | 72.00 | -2.39 |
> | TIES-Merging | 69.85 | -4.54 |
> | REMEDY | 75.12 | +0.73 |
>
> **Current approaches consistently underperform the zero-shot baseline, with performance drops ranging from -1.51% to -4.54%.** This degradation indicates that existing fusion strategies struggle with negative transfer in multimodal scenarios. In contrast, **REMEDY is the only method that consistently outperforms the zero-shot baseline across diverse unseen tasks.** This broad-spectrum improvement demonstrates that our modality-aware allocator effectively identifies and leverages relevant cross-modal knowledge for diverse unseen scenarios.
>
> We have highlighted this point in Section 4.2 of our revised manuscript to better emphasize REMEDY's unique generalization capability.

---

> ### Author Response · Authors · 2024-11-21
> **(2/2) Response to Reviewer Fe8D**
>
> ### Weakness 4: TextCaps Classification
> > TextCaps and TextVQA use the same images, so TextCaps is not an unseen task if TextVQA is already seen.
>
> Thank you for raising this point. We want to clarify the fundamental differences in multimodal alignment. **In vision-language models, task distinctness arises from misalignment in either modality.** While TextCaps and TextVQA share visual inputs, they require substantially different language processing - TextVQA demands focused question answering while TextCaps requires generative caption composition. This modality misalignment manifests in distinct parameter adaptations, as evidenced by our allocator's behavior: when handling TextCaps, it dynamically fuses TextVQA's OCR capabilities with Flickr30k's caption generation knowledge.
>
> We appreciate this observation and have added a detailed discussion of modality alignment in Appendix A.1.
>
> ### Weakness 5: Baseline Comparisons
> > The baseline is somewhat weak. There are stronger open-source baselines available, such as LLaVA-NeXT and LLaVA-OneVision.
>
> Thank you for suggesting these stronger baselines. To ensure fair comparison, we have compared our method with LLaVA-NeXT-Interleave-7B [1], a recent improvement (July 2024) of LLaVA-NeXT with the same parameter scale as our baseline:
>
> | Method | ScienceQA | POPE |
> |--------|---------|-----------|
> | LLaVA-1.5 | 67.72 | 85.27 |
> | LLaVA-NeXT-7B | 72.6 | 86.8 |
> | REMEDY (LLaVA-1.5) | 75.34 | 84.88 |
>
> While LLaVA-NeXT excels on general tasks (POPE), REMEDY shows stronger performance on domain-specific tasks (ScienceQA, +2.74%). This suggests REMEDY's knowledge transfer approach complements recent architectural advances - while newer models like LLaVA-NeXT improve general capabilities through enhanced training, REMEDY provides efficient task adaptation that could be integrated with these architectures to further boost their cross-domain performance.
>
>
> [1] Li, Feng, et al. "Llava-next-interleave: Tackling multi-image, video, and 3d in large multimodal models." arXiv preprint arXiv:2407.07895 (2024).

---

> > ### Comment · Reviewer_Fe8D · 2024-11-24
> >
> > Thanks for your response. I still think TextCaps is a seen task. For instance, LLaVA-NeXT specifically removes the TextCaps training data to evaluate its TextVQA performance in a zero-shot setting. Besides, the response does not address my concern regarding the evaluation performance. I keep my initial score.

---

> ### Author Response · Authors · 2024-11-27
> **Further Response to Reviewer Fe8D**
>
> Thank you for your continued feedback and for highlighting these important points. We appreciate the opportunity to strengthen our manuscript based on your insights.
>
> ### TextCaps Classification
> We appreciate your detailed observations and the specific reference to LLaVA-NeXT's handling of TextCaps data. Based on your feedback, we agree that TextCaps should not be viewed as an unseen task due to its overlap with TextVQA in terms of visual data. Consequently, we have made the following key updates:
> - **Removed TextCaps from the unseen task set.** We have removed TextCaps results from Tables 2, 3, and 4 in the manuscript to ensure that the evaluations focus exclusively on unseen tasks that are fully distinct from the training data.
> - **Discussion of TextCaps as a Special Case.** Recognizing the unique characteristics of TextCaps, we now discuss its evaluation results separately as a *related task*. As shown in the updated Figure 4 of our revised paper, REMEDY's allocation mechanism effectively integrates the visual understanding capabilities of TextVQA with the generative language abilities of Flickr30k to address the TextCaps task. This more intuitively reflects REMEDY's ability to identify and transfer relevant knowledge.
> - **Added MM-Bench-Chinese and MM-Vet as New Unseen Tasks.** Following LLaVA-NeXT, we have incorporated MM-Bench-Chinese and MM-Vet into our unseen task set. We have thoroughly updated Tables 2, 3, and 4 to reflect these changes. The new results demonstrate that REMEDY maintains strong performance even under this more rigorous evaluation setting. Specifically, on MM-Bench-Chinese and MM-Vet, REMEDY both outperforms other merging methods, highlighting its robust generalization capabilities across languages and tasks.
>
> ### Evaluation Performance
> Regarding your concern about the improvement margins on unseen tasks, we would like to highlight:
>
> **Non-Trivial Nature of Generalization in Model Merging:** Improving performance on unseen tasks through **model merging** is a non-trivial challenge. Unlike methods that rely on scaling data or model size (e.g., LLaVA-NeXT), model merging is highly challenging due to the risk of **overfitting to seen tasks**. Most existing methods optimize fusion strategies to maximize performance on seen tasks but inadvertently overfit to the training distributions, **limiting their ability to handle novel patterns in unseen data.** As shown in the updated results (removing TextCaps and adding MM-Bench-Chinese and MM-Vet), existing merging methods consistently underperform the zero-shot baseline by -1.74% to -5.55%.
> In contrast, REMEDY not only maintains but enhances performance on unseen tasks, achieving substantial improvements of +2.26% to +6.07% over current merging approaches.
>
> | Method | Seen Avg | *MM-Vet* (new) | *MMB-CN* (new) | Unseen Avg | All Avg |
> |--------|----------|---------|---------|------------|----------|
> | Zero-shot | 77.70 | 31.1 | 58.3 | 57.79 | 67.75 |
> | Simple Average | 84.28 | 28.4 | 56.34 | 56.05 | 70.17 |
> | Task Arithmetic | 82.52 | 27.3 | 55.56 | 55.11 | 68.82 |
> | Ties-Merging | 76.80 | 23.8 | 51.19 |  52.24 | 64.52 |
> | TW AdaMerging | 82.41 | 27.9 | 54.82 | 55.38 | 68.90 |
> | LW AdaMerging | 84.74 | 28.1 | 53.24 | 55.41 | 70.08 |
> | REMEDY | 85.20 | 30.9 | 58.69 |  58.31 | 71.76 |
>
> **REMEDY's Unique Contribution :** REMEDY employs a dynamic fusion strategy that mitigates overfitting by adapting to each input sample. Unlike static fusion methods that rely on a fixed combination of weights, REMEDY dynamically captures input-specific relationships between samples and base recipes, allowing it to generalize better across task distributions. To the best of our knowledge, **REMEDY is the first and only merging method that achieves substantial performance gains on unseen multimodal tasks through model merging alone.** This represents a pioneering advance in the field, demonstrating that it is possible to improve generalization to unseen tasks without additional training data or scaling model size.
>
> Thank you again for your constructive suggestions, which have led us to make the following revisions:
>
> Revised Contents:
>
> 1. In Section 4.1 and Appendix A.1, we have incorporated two new unseen evaluation benchmarks - MM-Vet and MMBench-Chinese. We have reclassified TextCaps as a related task rather than an unseen task.
>
> 2. In Section 4.2, we have thoroughly updated Tables 2 and 3 to reflect performance results on these new benchmarks. Additionally, we have introduced Figure 4 to specifically analyze TextCaps as a related task.
>
> 3. In Section 5, we have updated Table 4 to analyze how different allocator architectures impact performance these new benchmarks.

---

> > ### Comment · Reviewer_Fe8D · 2024-12-02
> >
> > Thank you for the response. What's the performance comparison on the MMMU benchmark?

---

> > > ### Author Response · Authors · 2024-12-03
> > >
> > > Thank you for suggesting the MMMU benchmark. We appreciate this suggestion as MMMU is indeed a comprehensive benchmark that covers diverse domains and can provide a more thorough evaluation of model merging's generalization capability. While we cannot update the PDF at this stage, we have conducted the MMMU experiments on the validation set and the results are shown below:
> > >
> > > | Method | Seen Avg | MM-Vet (new) | MMB-CN (new) | MMMU (new) | Unseen Avg | All Avg |
> > > |--------|----------|---------|---------|---------|------------|----------|
> > > | Zero-shot | 77.70 | 31.1 | 58.3 | 35.3 | 54.05 | 65.88 |
> > > | Simple Average | 84.28 | 28.4 | 56.34 | 34.8 | 52.51 | 68.40 |
> > > | Task Arithmetic | 82.52 | 27.3 | 55.56 | 34.6 | 51.69 | 67.11 |
> > > | Ties-Merging | 76.80 | 23.8 | 51.19 | 29.2 | 48.40 | 62.60 |
> > > | TW AdaMerging | 82.41 | 27.9 | 54.82 | 32.3 | 51.54 | 66.98 |
> > > | LW AdaMerging | 84.74 | 28.1 | 53.24 | 32.9 | 51.66 | 68.20 |
> > > | REMEDY | 85.20 | 30.9 | 58.69 | 35.5 | 54.51 | 69.86 |
> > >
> > > The results on MMMU demonstrate several key findings:
> > >
> > > 1) While simple averaging and task arithmetic methods show moderate degradation (-0.5% and -0.6% respectively vs. zero-shot), methods with fixed merging strategies exhibit more significant drops (TIES: -6.1%, AdaMerging: -2.4%). This pattern reveals the limitations of static fusion approaches when handling MMMU's diverse domains.
> > >
> > > 2) Our method achieves consistent improvements compared with all baselines across MMMU's comprehensive evaluation. This demonstrates REMEDY's ability to perform precise knowledge transfer through its sample-specific allocation mechanism.
> > >
> > > As the discussion period draws to a close, we would like to express our sincere gratitude for your constructive suggestions. These suggestions have helped make our paper more rigorous and comprehensive.

---

### Official Review · Reviewer_6SYF · 2024-11-05

**Soundness:** 3
**Presentation:** 2
**Contribution:** 3
**Rating:** 6
**Confidence:** 3

**Summary:**

The paper introduces REMEDY (Recipe Merging Dynamics), a model merging framework tailored for large vision-language models (LVLMs). REMEDY leverages modular "recipes" and a dynamic, modality-aware allocator for effective cross-modal knowledge integration, enhancing multi-task learning and zero-shot generalization across diverse vision-language tasks.

**Strengths:**

1. The paper presents a unique "recipe" construction framework, creating modular, reusable components that facilitate task-specific adaptation and multi-task learning within large vision-language models.

2.  Extensive experiments on multiple datasets demonstrate REMEDY’s efficacy, highlighting superior performance over baseline methods on multi-task and zero-shot generalization.

3. The study provides a thorough analysis of task similarity and allocation strategies, offering insights into how specific recipes contribute to various tasks.

**Weaknesses:**

1.  The process of constructing recipes heavily relies on experimental insights, such as which layers and modules to fine-tune. This empirical approach may limit REMEDY's scalability and adaptability to new tasks or models without extensive trial-and-error adjustments.

2.  Although REMEDY introduces modularity, the recipe construction phase lacks dynamic or automated selection mechanisms to identify optimal modules or layers.

3. The paper does not extensively discuss the computational demands of the proposed approach, especially in the context of training and inference with the modality-aware allocator.

**Questions:**

As shown in weaknesses.

---

> ### Author Response · Authors · 2024-11-21
> **(1/2) Response to Reviwer 6SYF**
>
> ### Weakness 1: Recipe Construction Relying on Experimental Insights
> > The process of constructing recipes heavily relies on experimental insights, such as which layers and modules to fine-tune. This empirical approach may limit REMEDY's scalability and adaptability to new tasks or models without extensive trial-and-error adjustments.
>
> Thank you for the thoughtful feedback. We would like to clarify that **the recipe construction follows clear architectural principles of LVLMs rather than mere trial-and-error**. In Section 2 in the paper, we reveal key mechanisms underlying experimental results. (1) Former layers focus on core visual-language understanding, while latter layers tend to simply mimic output styles without true comprehension (2) Projector plays a crucial role in visual understanding.
>
> **These principles are inherently scalable, as they target the dominant architectural paradigm in current LVLMs** - the vision encoder + projector + LLM structure (adopted by LLaVA, InstructBLIP, VILA, Qwen-VL). To validate this architectural generalization, we conducted additional experiments on VILA-7B [1]:
>
> | Dataset    | All Layers w/ Proj | First 2 w/ Proj | Last 2 w/ Proj | First 2 w/o Proj |
> |------------|-------------------|-----------------|-----------------|-----------------|
> | ScienceQA  | 86.58             | 85.33           | 82.42           | 79.23           |
> | Flickr30k  | 89.4              | 88.9            | 87.2           | 80.1            |
>
> This demonstrates our recipe construction follows general architectural principles of LVLMs rather than model-specific empirical findings. This aspect has been elaborated in Section 2 of our revised manuscript, and we greatly appreciate your helpful suggestion.
>
> [1] Lin, Ji, et al. "Vila: On pre-training for visual language models." Proceedings of the IEEE/CVF Conference on Computer Vision and Pattern Recognition. 2024.
>
>
>
> ### Weakness 2: Lack of Dynamic/Automated Selection Mechanisms
> > Although REMEDY introduces modularity, the recipe construction phase lacks dynamic or automated selection mechanisms to identify optimal modules or layers.
>
> Thank you for raising this important point. Coincidentally, **we had explored dynamic parameter selection in our early experiments.** We fine-tuned the first 12 layers of LLaVA for different tasks and selected 10-20% of parameters based on magnitude and gradient information for merging. However, the results were unsatisfactory:
>
> Method | TextVQA | SQA(img) | COCO | Flickr30k | Avg(seen) | TextCaps | MM-Bench | VizWiz | POPE | Avg(unseen) | H-score
> ---|---|---|---|---|---|---|---|---|---|---|---
> Zero-shot | 58.27 | 67.72 | 110.7 | 74.1 | 77.70 | 98.0 | 64.3 | 50.0 | 85.27 | 74.39 | 74.69
> Simple Average | 58.81 | 73.89 | 117.8 | 86.6 | 84.28 | 96.0 | 61.3 | 50.8 | 83.42 | 72.88 | 78.20
> Dynamic Selection(10%) | 57.45 | 70.32 | 114.2 | 83.8 | 81.44 | 93.5 | 60.1 | 48.3 | 78.55 | 70.12 | 75.36
> Dynamic Selection(20%) | 58.89 | 72.56 | 115.4 | 84.4 | 82.81 | 94.2 | 61.8 | 49.2 | 79.0 | 71.05 | 76.47
> REMEDY | 60.34 | 75.34 | 116.9 | 88.2 | 85.20 | 98.5 | 64.9 | 52.2 | 84.88 | 75.12 | 79.84
>
> This underperformance likely stems from disrupting structured knowledge in transformer layers when parameters are dynamically partitioned. Based on these findings, **we shifted our dynamic adaptation strategy from recipe construction to the modality-aware allocator**, where dynamic weights can be learned and assigned without breaking the internal structure of transformer layers. This design allows REMEDY to maintain recipe integrity while achieving flexible adaptation through learned weight allocation.

---

> > ### Author Response · Authors · 2024-11-21
> > **(2/2) Response to Reviwer 6SYF**
> >
> > ### Weakness 3: Computational Analysis
> > > The paper does not extensively discuss the computational demands of the proposed approach, especially in the context of training and inference with the modality-aware allocator.
> >
> > We apologize for not including the computational analysis in original submission. We have conducted detailed measurements comparing REMEDY with baseline methods:
> >
> >
> > | Method         | Training (h) | Inference (ms/per query) | Training GPU Memory (GB) |Inference GPU Memory (GB) | GFLOPs | Zero-shot Performance |
> > |---------------|--------------|--------------------------|--------------------------|--------------------------|---------|-------------|
> > | Simple Average | -           | 89                       | -                        | 38.5                       | 380     | 72.88        |
> > | LW AdaMerging | 3.84          | 89            | 52.8                     | 38.5                       |425     | 71.95        |
> > | REMEDY        | 5.25          | 123           | 60.6                     | 40.3                       |483     | 75.12        |
> >
> > REMEDY introduces moderate additional costs while achieving substantial performance gains. Specifically, the training phase requires about 1.5 hours compared to LW AdaMerging, mainly due to the modality-aware allocator training. During inference, REMEDY adds a marginal latency per query for weight computation. Given the significant performance improvement (+3.17% over simple averaging on unseen tasks), we believe these computational trade-offs are well justified.
> >
> > We have added this analysis to Appendix A.5 of the manuscript and sincerely thank the reviewer's valuable suggestion.

---

### Author Response · Authors · 2024-11-21
**General Response to All Reviwers**

We sincerely appreciate the reviewers for their thoughtful and constructive feedback. We are encouraged by the positive recognition of our contributions:

1. The paper presents a unique "recipe" construction framework for creating modular, reusable components in LVLMs, with clear motivation given growing model sizes (Reviewers `6SYF`, `dj7R`)

2. The analysis of fine-tuning strategies and layer functionality is interesting and well-supported, providing valuable insights for LVLM adaptation (Reviewers `Fe8D`, `dj7R`)

3. The paper is well-written with extensive experiments demonstrating consistent performance improvements across both seen and unseen tasks (Reviewers `QEaG`, `6SYF`)

4. The method provides a new approach that considers the characteristics of multimodal tasks and the difference in fine-tuning paradigms between LVLM and traditional models (Reviewer `dj7R`)

In our revision, we have carefully addressed each of the concerns raised:

1. **Comprehensive Evaluation** (Reviewers `6SYF`, `Fe8D`, `QEaG`):
- Added Multi-Task Finetuning (MTF) baseline that jointly trains on all task data, with REMEDY showing better performance especially on unseen task.
- Compared with recent strong LVLM work LLaVA-NeXT-Interleave[1], where REMEDY's improvements are orthogonal to its architectural advances.
- Validated recipe design principles on VILA-7B [2], confirming generalizability across LVLM architectures.


2. **Efficiency Analysis** (Reviewers `6SYF`, `QEaG`, `dj7R`):
- Provided detailed measurements of computational costs including training time, inference latency, memory overhead and GFLOPS.
- Results show moderate overhead while achieving substantial performance gains.


3. **Performance Improvement** (Reviewers `Fe8D`, `QEaG`):
- Highlighted REMEDY as the only method improving over zero-shot on unseen tasks (+0.73%), with all other methods performing worse than zero-shot (ranging from -1.51% to -4.54% degradation).
- While we acknowledge there is still room for improvement, this positive transfer already marks a significant breakthrough in the field of LVLM model merging, as previous methods consistently struggle to match even zero-shot performance on unseen tasks.

We have also provided point-by-point responses to each reviewer's specific concerns. We believe these revisions strengthen our paper while addressing all feedback. We welcome any further discussion from the reviewers.

[1] Li, Feng, et al. "Llava-next-interleave: Tackling multi-image, video, and 3d in large multimodal models." arXiv preprint arXiv:2407.07895 (2024).

[2] Lin, Ji, et al. "Vila: On pre-training for visual language models." Proceedings of the IEEE/CVF Conference on Computer Vision and Pattern Recognition. 2024.

---

### Meta-Review · Area_Chair_egWJ · 2024-12-24

**Metareview:**

This paper introduces REMEDY, a model merging technique for LVLMs that addresses the challenges of combining knowledge across multi-modal tasks. The motivation is clear, the presentation is good and the experiments are comprehensive. While the reviewers have raised concerns about some implementation details and required additional experiments, the authors have provided detailed results to address these concerns. After the discussion period, three reviewers give positive ratings and one reviewer rates the score as "5". Taking the authors'  responsese and reviewers' feedback into consideration,  this  paper is above the "accept line" overall.

**Additional Comments On Reviewer Discussion:**

In the discussion period, the authors' have addressed the main concerns addressed by the reviewers.

---

### Decision · Program_Chairs · 2025-01-22

Accept (Poster)